# Kinetics of Resorcinol-Formaldehyde Condensation—Comparison of Common Experimental Techniques

**DOI:** 10.3390/gels8010008

**Published:** 2021-12-23

**Authors:** Eva Kinnertová, Václav Slovák, Roman Maršálek, Martin Mucha

**Affiliations:** Department of Chemistry, Faculty of Science, University of Ostrava, 30. dubna 22, 70103 Ostrava, Czech Republic; vaclav.slovak@osu.cz (V.S.); roman.marsalek@osu.cz (R.M.); martin.mucha@osu.cz (M.M.)

**Keywords:** resorcinol-formaldehyde, sol-gel kinetics, DSC, ^1^H-NMR relaxometry, FTIR, DLS

## Abstract

Porous carbons, originated from resorcinol-formaldehyde (RF) gels, show high application potential. However, the kinetics and mechanism of RF condensation are still not well described. In this work, different methods (dynamic light scattering–DLS, Fourier transform infrared spectroscopy–FTIR, low field ^1^H nuclear magnetic resonance relaxometry–^1^H-NMR, and differential scanning calorimetry–DSC) were used to follow the isothermal RF condensation of mixtures varying in catalyst content (Na_2_CO_3_) and reactant concentration. The applicability and results obtained by the methods used differ significantly. The changes in functional groups can be followed by FTIR only at very early stages of the reaction. DLS enables the estimate of the growth of particles in reaction solution, but only before the solution becomes more viscous. Following the relaxation of ^1^H nuclei in water during RF condensation brings a different view on the system—this technique follows the properties of the present water that is gradually captured in polymeric gel. From this side, the process behaves similarly to the nucleation reaction, which is in contradiction to the n-order mechanism confirmed by other techniques. The widest range of applicability was found for DSC measurement of the freezing/melting behavior of the reaction mixture, which is possible to use without any limitations until full solidification. Furthermore, this approach enables us to follow the gradual formation and development of the gel through the intermediate undergoing glass transition.

## 1. Introduction

Resorcinol (R) and formaldehyde (F) are precursors for the preparation of highly porous and chemically resistant carbon materials [1]. The exceptional properties predetermine their use as adsorbents [2], catalyst supports [3], electrodes for supercapacitors, batteries or fuel cells [4,5,6,7], and column packing materials for chromatography [8]. The preparation of these materials is based on the sol-gel polycondensation of the precursors (R and F) under alkaline or acidic conditions [9].

Over the last 30 years, a large number of investigations have been performed focusing on the study of the properties of RF gels. Six hundred and sixteen studies are found on the Web of Science that have in the title concept *resorcinol-formaldehyde*, and only 15 of them contain *kinetic* in the title. Nevertheless, most of these ‘kinetic’ papers follow the kinetics of adsorption in RF-based carbon adsorbents. In fact, only five contributions have attempted to study and explain the course and mechanism of the kinetics of resorcinol-formaldehyde polycondensation. This lack of mechanistic or kinetic studies is in clear contradiction with the generally adopted fact that RF condensation is the crucial step of RF based carbons. Therefore, its understanding is necessary for the effective tailoring of the properties of final gels.

The probable reason why this issue is relatively poorly explored is in technical matters. Moreover, only a limited number of techniques allow for the investigation of the whole course of the sol-gel reaction. To a lesser degree, the problem is not at the beginning, when the reaction starts from the RF polycondensation mixture (sol), which can be examined by various techniques. However, the progress of the restriction comes with the reaction progress when the sol develops through various polymeric intermediates into a three-dimensional polymer network (gel) [10].

The structure formation during sol-gel transition of the RF solution has been the most investigated by light scattering methods, dynamic light scattering (DLS), and static light scattering (SLS). A significant number of researches have been carried out in this area [11,12,13,14,15]. According to Yamamoto et al. [13], the growth rate of primary clusters in the early gelation stage was particularly dependent on the catalyst concentration (R/C molar ratio). On the contrary, Gaca and Sefcik [14] concluded that the size of primary clusters in the early gelation stages was independent of the R/C ratio and the reactant concentration (R and F). However, a higher catalyst concentration led to an increase in the number of clusters. Moreover, Taylor et al. [15] investigated the kinetics of cluster growth with the use of various carbonates as catalysts and reached similar findings as the previous researchers [14].

The second most used technique for investigating sol-gel RF polycondensation is nuclear magnetic resonance spectroscopy (^1^H NMR or ^13^C NMR) [16,17]. Gaca et al. [18] studied the kinetics of reaction between R and F under alkaline conditions (Na_2_CO_3_) by ^1^H NMR and ^13^C NMR. The authors not only confirmed the presence of substituted species of resorcinol, but also revealed that a significant interaction of reactants (R and F) took place almost immediately after mixing resorcinol with formaldehyde in the presence of catalyst. Lewicki et al. [19] investigated the mechanism and kinetics of the formation of the RF network by combination ^1^H NMR, low field NMR relaxometry, and differential scanning calorimetry (DSC). In this study, DSC was used only as a tool for proving the exothermic character of the reaction. In addition, the kinetic evaluation was predominantly based on the treatment of the relaxometry data (relaxation time), which revealed that polycondensation follows first-order kinetics.

Due to the exothermic character of resorcinol-formaldehyde condensation, it can be successfully followed by DSC. Šebenik et al. [20] were probably the first researchers which studied this complex reaction between R and F in the presence of NaOH (as a catalyst) using DSC with a combination of high performance liquid chromatography (HPLC), ^1^H NMR, ^13^C NMR, and infrared spectroscopy (IR). They identified substituted resorcinol species and found that the first step corresponds to the second order kinetics and the second step corresponds to the first order reaction model. Recently, our team has studied the kinetics of the reaction of R with F during heating (different heating rates) by DSC and we confirmed the second order kinetic model for the first reaction step. However, the second step was better characterized by the contracting sphere model [21].

Information regarding changes in the composition of the reaction mixture can also be provided by Fourier transform infrared spectroscopy (FTIR). Lian et al. [22] determined the free formaldehyde content during polycondensation of phenol-formaldehyde mixtures by near infrared spectroscopy (FT-NIR). FTIR was also used to prove the creation of cross-linkage in RF gels. However, functional groups were investigated on synthetized gels only [23,24]. In general, FTIR is used mainly to examine the functional groups of the final organic and carbon RF gels in the available literature [25,26,27,28].

The aim of this contribution was to revisit the field of RF condensation mechanism and kinetics with various widely available techniques (DLS, FTIR, ^1^H-NMR relaxometry, and DSC) and to demonstrate their strengths and/or weaknesses together with the range of their applicability. With all the methods, well-defined reaction systems with various catalyst concentrations (R/C molar ratio 25 and 50) and the concentration of reactants (20 and 40 weight %) were observed isothermally at 40 °C from the start of the reaction until full gelation.

## 2. Results

### 2.1. DLS

The DLS method is based on measuring the intensity of scattered light. Scattering occurs upon contact of a beam of radiation on the particles. Therefore, it is clear that the intensity of the scattered light will increase with the amount and size of these particles in suspension. The intensity of scattered light can be expressed using the mean count rate (Figure 1).

Regarding the three studied systems, the intensity of scattered light gradually increases, indicating the formation of clusters. The rate of this process depends on the composition of the reaction mixtures. The fastest increase in the intensity of scattered light was recorded for the RC25-40 sample, which represents a higher catalyst content and a higher R + F concentration. The decrease in catalyst content (sample RC50-40) and the concentration of reactants (RC25-20) lead to a significant decrease in the recorded intensity of scattered light.

Using the autocorrelation function, it is possible to calculate the hydrodynamic radius of the present particles, respectively their average value D_ave_ (Figure 2).

Regarding the three samples, the smallest detectable particle size is around 2 nm. The effect of catalyst amount (RC25-40 versus RC50-40) is negligible at the early stage of reaction (up to 50 min), which is consistent with the findings of other authors [14]. On the contrary, these samples significantly differ in the intensity of scattered light (Figure 1). Therefore, the amount of catalyst affects the number of growing particles and not their size (at least at the start of the reaction). The deviation of the last two experimental points of RC25-40 (the fastest reaction) to the higher particle diameter does not necessarily need to be related to the increase in particle size, due to the rapidly increasing viscosity of this mixture, which causes lower particle mobility and overestimation of particle size.

### 2.2. FTIR

The continuous recording of the FTIR spectra of the studied reaction mixtures showed (example in Appendix A) that changes in the spectra can be observed only in the first hour after mixing the precursors. Later, the differences are negligible.

The main changes in the spectra occur in the region of C=C stretch vibrations (1650–1450 cm^−1^) and in the region of C–O stretch and C–H deformation vibrations (1200–950 cm^−1^). Slight shifts of the bands occur in the case of C=C stretch vibrations, which could be assigned to the crosslinking among the resorcinol and formaldehyde, which leads to the changes caused by mesomeric or steric effects. Changes in the region 1200–900 cm^−1^ can be explained by an interaction between resorcinol hydroxyl groups with formaldehyde molecules. Various compounds containing oxygen can be formed during the initial phase of the reaction. The C=O stretch vibration band is not observable in the spectra since it is hidden in the broad deformation band of water [29].

Time dependences of the band absorbances are shown in Figure 3. The decrease of absorbance of bands at 1151 cm^−1^ (Figure 3a), 1057 cm^−1^ (Figure 3c), and 964 cm^−1^ (Figure 3d) during the reaction can be observed. Bands 1151 and 1057 cm^−1^ can be assigned to C–O stretch vibrations in resorcinol, and the band at 964 cm^−1^ probably belongs to the C–H deformation vibration in the formaldehyde. The increase in band absorbance at 1114 cm^−1^ (Figure 3b) can be assigned to the formation of acetals, which can act as a crosslinking agent in the emerging material. Changes in the spectra can be easily observed in the case of mixtures RC50-40 and RC25-40, but they are small in the case of RC25-20, which is caused by different concentrations of precursors. The concentration is significantly higher for the mixtures RC50-40 and RC25-40 compared to RC25-20. The rate of changes is significantly higher for the RC25-40 mixture, as it contains the higher amount of catalyst compared to the RC50-40. The absolute change in the absorbances of all the observed bands is also higher in the case of RC25-40.

### 2.3. NMR

With the NMR relaxation technique, we follow the relaxation of ^1^H nuclei. Although the measured mixtures contain up to 40 mass % of RF precursors, water represents more than 90% of the present molecules, and with the reaction progress this portion increases. On this basis, we expect that the measured signal (decreasing magnetization) is related mainly to the relaxation of hydrogen nuclei of water molecules (in which we will simply use the term “relaxation of water” in the following text).

In general, the rate of relaxation of liquid water (expressed here as relaxation time) depends on several factors (concentration and character of dissolved solids, presence of paramagnetic centers, possibility of free movement, mobility of molecules, etc.). In this complex system with the reacting RF mixture studied in this work, a continuous distribution of water molecules varies in relaxation times from very fast relaxation below 1 ms up to slowly relaxing bulk water (with typical relaxation time about 3500 ms). Then, the experimental relaxation curve (decrease of magnetization with time) reflects this continuous distribution. In our case, the preliminary analysis of relaxation curves obtained for the studied RF mixtures during various stages of condensation showed us clearly that the observed relaxation curves can be very well approximated, expecting only two kinds of water molecules with different relaxation times. The addition of another component did not bring any statistically significant improvement.

As a result, for the further evaluation, only two components in Equation (4) were expected, indicating that two “types” of relaxing water molecules are present. The evolution of relaxation times T_2_ of both components with the reaction time is depicted in Figure 4, while their abundance is illustrated in Figure 5.

Although both of the expected components have to be understood as formal only, their properties (relaxation time and abundance, Figure 4 and Figure 5) enable us to assign them to states of water in the studied system. The component 1 (C1 in the next text), with typically longer relaxation and higher content at the start of the reaction, could represent bulk water with more or less free mobility. The component 2 (C2), with opposite properties to C1, could be related to water molecules in close proximity to the emerging particles, with a limited possibility of motion, and thus a shorter relaxation time.

The general trends measured for all the studied mixtures are practically the same and differ only in the time at which the observed changes occur. At the start of the reaction, C1 predominates (Figure 5) with the relaxation times related to the concentration of reactants. In a more diluted solution, RC25-20 relaxes almost 3000 ms, and the more concentrated RC25-40 and RC50-40 relax faster (1800 ms). With the continuous condensation and bonding of additional reactants to initial clusters, the solutions become more diluted, and the relaxation of C1 is prolonged and reaches the maximum (Figure 4). Then, the subsequent growth of clusters starts to limit the mobility of C1 water molecules and the relaxation time has a decreasing trend until the minimum. This minimum at the dependence of T_2_ on the reaction time coincides with the dramatic change in the proportion of C1 and C2 (inflex points in Figure 5). At this time (18, 25, and 53 min for RC25-40, RC-50-40, and RC25-20, respectively), C1 is quickly transformed to C2 with a considerably shorter relaxation time. This transformation is probably related to the size of present particles. Here, the observed times correspond to the hydrodynamic diameter by about 5 nm for all the mixtures (see Figure 2). Therefore, the particles of this size seem to be capable of affecting the solution in the whole volume, radically decreasing the mobility of water molecules, and dropping down the observed relaxation times, which we detect as a fast increase of the formal component C2 amount. Of note, in this critical moment the reaction mixtures are still liquid, although the increased viscosity could be expected. Following the continuation of the reaction does not bring any significant changes of C1 and C2 abundance. In this case, C2 predominates and its relaxation times consequently decrease to very low values (units of milliseconds), close to the relaxation of ^1^H nuclei in a solid state.

### 2.4. DSC

The freezing/melting behavior of all the mixtures during the reaction was followed by DSC. While the freezing of the reaction mixture was practically unaffected by the progress of the reaction, two variants of melting curves were observed (Figure 6). In addition to the simple endothermic peak related to melting and observable at the start and end of reaction, in a specific time regime a more complicated process was observed. In the latter case, a considerable exothermic peak was observed at low temperatures, only at the beginning of the melting process. The effect is clearly observable for RC25-40 (between 67 and 120 min of reaction), less distinct for RC50-40 (174–335 min), but could not be observed for RC25-20. To explain this phenomenon, the heat evolved during freezing or consumed for melting was evaluated from the area of peaks at the DSC curves (Figure 7).

The trends (in heat) are the same for freezing and melting. The heat of fusion increases in the early stage of the reaction due to the increasing content of water produced by the polycondensation reaction. Then, for RC25-20, the heat remains practically constant. For the other two mixtures, the heat starts to decrease from the specific moment (approximately in 70 and 150 min for RC25-40 and RC50-40, respectively). It is noticeable that this phenomenon starts in the reaction phase when the samples are fully solidified (see the results of tilt tests in the Section 5), although not fully cured/cross-linked. At this stage, we expect that the mixture contains isolated 3D macromolecules swelled with water in their structural voids of subnanometric size. During freezing, the polymer through glass transition becomes much more rigid and traps the water in the form of nanodroplets which are not able to freeze (the presence of non-freezing water in porous systems is well known), and the heat of fusion of this system is lowered. During consequent heating, when reaching the glass transition temperature of the present polymer, the structure becomes more flexible, the water is released, and ice crystals can form (observed exothermic effects, Figure 6). At higher temperature, the water simply melts. A similar behavior was described for polysaccharide solutions [30,31,32]. To the best of our knowledge, it has not been reported for RF polymers yet.

When the reaction continues, macromolecules become interconnected, the structure is more rigid even at room temperature, and the water captured remains unfrozen during the measurement cycle—the exothermic effect disappears, and the heat of fusion becomes constant.

The described phenomenon was not observed for the RC25-20 mixture, probably since the solution is diluted and the effect is too small (if any) to be detectable.

In Figure 6, it can be seen that the melting peak temperature gradually increases with the reaction progress, which is clearly illustrated in Figure 8.

At the start of the reaction, the melting temperature corresponds to the aqueous solution of R, F, and catalyst. However, the first intermediates (soluble low molecular weight oligomers) formed immediately after mixing the reactants [18]. Therefore, it is understandable that the melting temperature of the more diluted RC25-20 mixture starts at a higher level compared to the highly concentrated ones. Moreover, it is expected that the melting of RC25-40 starts at a higher temperature than RC50-40, since RC25-40 reacts faster and more intermediates can be formed (leading to a more diluted solution) before the first measurement.

With the increasing time of reaction, the melting temperature increases in all of the studied cases to a more or less constant value at a later stage of reaction. The rate of increase is in a non-surprising order RC25-40 > RC50-40 > RC25-20. On the other hand, surprisingly, the final melting temperature of the RC50-40 sample (approximately −8 °C) is considerably lower than the samples with a higher amount of catalyst (−2 °C), which was confirmed by several repetitions of the experiment. Hypothetically, the reason may be a higher residual concentration of reactants in the aqueous component of the final gel. A lower concentration of catalyst can lead to a lower number of initial clusters and bigger particles (which agrees with our findings from DLS measurements and with the literature, e.g., [33]), lower consumption of the reactants during their growth, and consequently a higher residual concentration of reactants in the aqueous phase after gelation/solidification.

## 3. Discussion

### 3.1. Kinetic Analysis

The kinetic analysis (described in the Section 5) leads to a set of kinetic parameters based on FTIR, NMR relaxometry, and DSC measurements (Table 1).

The results show that all of the methods are able to describe similar trends in rate constants, showing higher values for RC25-40 (higher catalyst content and higher concentration of reactants) and significantly lower values, but close to each other, for samples with a decreased amount of catalyst (RC50-40) or concentration of reactants (RC25-20). Differences in the values of the rate constant obtained from different methods are understandable, since each of them focuses on principally different components of the reaction system. While FTIR is dependent on bonds and groups in the reacting system (for which the changes are considerable only at the early start of the reaction), NMR follows the relaxation (mobility) of water molecules affected by the presence of solid particles, and DSC measures the changes in the melting point of the complex mixture of water, reactants, and various intermediates, which are slower and detectable for a considerably longer time (thus leading to considerably lower values of rate constants).

Interesting conclusions can be based on the analysis of the obtained values of parameters *n* and *m*, which determine the dependence of the reaction rate on the conversion (the so-called reaction model). FTIR and DSC experiments lead to values of *n* between 1 and 1.6 and *m* close to zero. This indicates that the reaction followed by these techniques can be described with the model of kinetics of n-order with n > 1. It partly agrees with the data in the literature for the first reaction step of RF condensation (the second-order kinetic model was confirmed in [20,21]). On the other hand, the values of *n* and *m* obtained from the NMR experiments are both close to 1. These similar values of *n* and *m* are typical for the Avrami-Erofeyev reaction models [34] based on the nucleation and nuclei growth restricted by the elimination of a potential nucleation site by the growth of an existing nucleus and/or loss of the reactant/product interface. This occurs when the reaction zones of two or more growing nuclei merge [35]. It is not without meaning to expect this mechanism of growing particles (affecting water relaxation) during RF condensation.

The results clearly indicate that all of the three methods can be simply used for following RF condensation and studying the effect of various process parameters on its kinetics. Moreover, all of the three methods (especially DSC) can be easily adapted for the measurement at different temperatures, which opens the door for the estimation of the temperature function (temperature dependence of rate constant) and consequent kinetic modelling of the process.

### 3.2. Comparison of the Techniques

To compare the used techniques and illustrate their applicability, the dependence of conversion obtained from FTIR, NMR, and DSC (see the Section 5) or the hydrodynamic diameter of particles from DLS on time were depicted in Figure 9.

The applicability of FTIR for kinetic and mechanistic studies of RF condensation is limited to the early start of the reaction. Regarding all of the studied samples, the FTIR records have become constant after 1 h of measurement. Although this technique can also be used for kinetic studies, it can provide information only regarding the initial reaction step, in which considerable composition changes are detectable. On the other hand, FTIR is the only technique (used in this contribution) that provides direct information regarding real chemical reactions during RF condensation.

NMR relaxometry presents an interesting and not very often used method for studying RF condensation kinetics. The results mentioned above show that the nucleation and growth of particles (and not chemical changes) are well accessible by this method through their effect on the mobility of water molecules (related to measured relaxation). Moreover, the methodology used is limited to the start of the reaction, before the particle which is larger than about 5 nm appears in the system (compare with the DLS particle diameter in Figure 9). Here, it can be expected that the optimization of the experimental methodology (pulse sequence, data treatment) would extend the applicability of this method in the future.

DLS would provide information on the size and amount of particles in the solution. The measured signal (intensity of scattered light) keeps the increasing trend for a longer time than FTIR or NMR methods (for the studied samples). Its applicability is wider (Figure 9), but limited to a time regime in which the viscosity of solution (naturally increasing) can be expected as constant. Increasing the viscosity of the progress of the measured mixture with the reaction progress can affect the transformation of the measured quantity and leads to an overestimation of the particle size. It is not possible to simply convert DLS data to a conversion degree. Therefore, it is not suitable for the study of kinetics and mechanism of RF condensation.

Contrary to the previous methods, the continuous following of the freezing/melting behavior of the reaction mixture, combined with the appropriate isothermal steps at a given temperature with DSC, can be used from the start of the reaction in the liquid phase to the polymeric solid. The temperature of melting gradually increases and provides information regarding the system for a considerably longer time. At the same time, it enables us to determine the kinetic parameters of the reaction comparable with FTIR or NMR. In addition, these kinds of experiments allow us to detect this phase of reaction, in which polymeric structures undergo glass transition. This phase is closely related to the temperature of the observable gelation (Figure 9).

## 4. Conclusions

In this work, the kinetics and mechanism of RF condensation were investigated isothermally at 40 °C by DLS, FTIR, ^1^H-NMR relaxometry, and DSC. The studied reaction mixtures differed in the concentration of catalyst and reactants.

DLS provides information regarding the size and amount of particles in the solution, but only before the solution becomes more viscous. It was found that for all of the three mixtures, the particles size around 2 nm were detectable and the catalyst concentration did not affect the particles size, but rather, a number of growing particles in the reaction solution.

FTIR enables the investigation of the changes in functional groups at a very early stage of the reaction. Concurrently, it is the only one that provides direct information regarding real chemical reactions during condensation. The higher catalyst concentration led to a significantly higher rate of the observed changes.

Low-field ^1^H-NMR relaxometry allows for the investigation of the properties of present water, which is gradually captured in polymeric gel. Two types of water molecules were identified in the reaction systems—bulk water with more or less free mobility and water in close proximity to the emerging particles with a limited possibility of motion. The catalyst concentration only affected the time at which the observed changes occurred.

DSC is suitable for the examination of the reaction from the start in the liquid phase to the polymeric solid and enables the detection of the phase of reaction in which polymeric structures undergo glass transition, which is closely related to the temperature of the observable gelation. The higher catalyst concentration led to a faster reaction and the glass transition was observed in earlier times of reaction. The glass transition was not observed for a more diluted mixture (with a lower concentration of reactants).

Based on the FTIR and DSC experiments, the RF condensation can be described with the model of kinetics of n-order with n > 1. Concurrently, the mechanism of the growing particles was confirmed from the NMR experiments.

The combination of different techniques has helped in the better understanding of the course of a relatively complex RF condensation reaction.

## 5. Materials and Methods

### 5.1. Resorcinol-Formaldehyde Mixtures

The preparation of resorcinol-formaldehyde (RF) mixtures was based on the sol-gel polycondensation of resorcinol (≥99%, Carl Roth, Karlsruhe, Germany) and formaldehyde (38%, p.a., Mach Chemikálie s.r.o., Slezská Ostrava, Czech Republic) described elsewhere [36]. All of the RF mixtures were prepared using the molar ratio R/F = 0.5, with a different catalyst (C) concentration (molar ratio R/C = 25 and 50) and weight content of the reactants (w = 20% and 40%). In a typical procedure, 16.5 g of resorcinol was dissolved in 24.32 g of 38% aqueous formaldehyde. Then, 16 g of 2% (for R/C = 50) or 4% (R/C = 25) solution of Na_2_CO_3_ (anhydrous, Lachema) was added together with an appropriate amount of demineralized water. The examined polycondensation mixtures were labeled RCX-Y, where X represents the molar ratio of resorcinol to the catalyst and Y represents the weight content of the reactants. Polycondensation mixtures were prepared at room temperature (except for the case of NMR measurements, in which the polycondensation mixture was prepared from precursors preheated at 40 °C) and immediately investigated isothermally at 40 °C by low-field ^1^H-NMR relaxometry (NMR), differential scanning calorimetry (DSC), dynamic light scattering (DLS), and Fourier transform infrared spectroscopy (FTIR). If any pretreatment was needed before the specific technique, it is mentioned below. The time of the first measurement (at 40 °C) after mixing the reactants was up to 10 min for DLS and 5 min for the other techniques.

Regarding all of the studied mixtures, the simple tilt test was performed during heating in a thermostatic bath at 40 °C. The times of the start of solidification of the RC25-40, RC50-40, and RC25-20 mixtures were 74, 147, and 166 min, respectively.

### 5.2. DLS

The Zetasizer Nano-S system (Malvern Instruments Ltd., Malvern, UK) was used for the dynamic light-scattering measurements. The Zetasizer contains an avalanche photodiode detector (APD) along with a 4 mW He-Ne laser, which operates at a wavelength (*λ*) of 633 nm. The wave number of scattered light corresponding to the scattering angle of 173° is:(1)$$q=\frac{4\pi n}{\lambda }\sin\left(\frac{\theta }{2}\right)=0.019\;{\mathrm{nm}}^{-1}$$
where *n* is the refractive index of the solution [37].

The device measures the number of photons (kHz) incident on the detector (count rate). From the count rate value, it is possible to infer the presence of dust particles. However, it is also possible to deduce from the change of this quantity changes in the suspension, such as sedimentation or aggregation. An autocorrelation function is calculated from the primary signal, which indicates changes in suspension caused by the Brownian motion of the particles. The signal changes over time and small particles have a faster rate of change compared to the large particles.

When the autocorrelation decay showed an exponential decay, indicating unhindered Brownian diffusion, we used the cumulant method to estimate the initial decay rate *Γ* (s^−1^). From this, we can determine the mean diffusion coefficient *D*, using:(2)$$\Gamma =Dq^{2}$$
and the mean hydrodynamic radius *R_h_*, using the Stokes–Einstein equation:(3)$$D=\frac{k_{B}T}{6\pi \mu R_{h}}$$
where *k_B_* is the Boltzmann constant (J K^−1^), *T* is the absolute temperature (K), and μ is dynamic viscosity (mPa s) [14,15].

Moreover, the setup has a Peltier temperature controller, which allows for the change in the temperature from 2 to 92 °C.

The RF mixtures were filtered using a 0.22 µm pore size syringe filter (PTFE membrane) and 1 ml of the respective filtered mixture in a sealed polystyrene cuvette was equilibrated at 40 °C and analyzed in regular time intervals until solidification. All of the RF mixtures were measured twice. The time dependence was assessed for the mean count rate, correlation coefficient, and hydrodynamic radius.

### 5.3. FTIR

The measurement was carried out using the ATR technique on the Nicolet 6700 FTIR spectrometer (Thermo Scientific, Waltham, MA, USA). A single bounce diamond crystal was utilized for sample measurement, the resolution was set to 4 cm^−1^, and 16 scans were accumulated for each spectrum. The baseline correction was made prior to the interpretation.

The prepared mixtures sealed in glass tubes were placed in a thermostatic bath at 40 °C prior to the FTIR analysis. In regular time intervals (dependent on the rate of reaction), one drop of the mixture was taken for the analysis.

### 5.4. NMR

NMR measurements were performed using a low-field NMR spectrometer Minispec mq 20 (Bruker, Rheinstetten, Germany) operating at a proton resonance frequency of 20 MHz and a magnetic field strength of 0.47 T. Spin-spin relaxation T_2_ (transverse relaxation) of ^1^H was determined by the Carr-Purcell-Malboom-Gill (CPMG) pulse sequence with a pulse separation time of 90 and 180° of 1 ms. The total number of recorded echoes was 5000.

The polycondensation mixture was placed in the testing tube, capped, and placed in the preheated measurement chamber at 40 °C. The relaxation curves were recorded every 2 min.

We expected that the obtained dependence of magnetization on time (relaxation curve) is a linear combination of relaxation of two individual components differing in relaxation time *T*_2_ (e.g., hydrogen nuclei in bulk water and water trapped in the formed polymer structure). Therefore, the measured relaxation curve can be expressed as [38]:(4)$$M=M_{0}+A_{1}.e^{-\left(\frac{t}{T_{2-1}}\right)}+A_{2}.e^{-\left(\frac{t}{T_{2-2}}\right)},\;$$
where *M* is the measured magnetization in time *t*, *A*_1_, *A*_2_, and *T*_2−1_, *T*_2−2_ are the amplitudes and spin-spin relaxation times related to component 1 and 2.

Estimates of parameters *M*_0_, *A*_1_, *A*_2_, *T*_2−1_, *T*_2−2_ were found by biexponential fitting using the Scidavis software. The values of *A*_1_ and *A*_2_ were normalized after the calculation (*A*_1_ + *A*_2_ = 100).

### 5.5. DSC

DSC experiments were performed using DSC Q20 (TA Instruments, New Castle, DE, USA). An amount of 10 µl of the freshly prepared polycondensation mixture in hermetically sealed aluminum pans was equilibrated at 10 °C, and the first measurement cycle was applied. The measurement cycle consisted of cooling down to −50 °C (5 K min^−1^) and consecutive heating to 40 °C (5 K min^−1^). Between the measurement cycles, isothermal steps at 40 °C of 10 min length were applied (Appendix A). To place the DSC measurement results on an appropriate time scale comparable with the other techniques, the change in the extent of reaction (conversion) during the cooling and heating phases had to be estimated. Based on the kinetic equation published earlier for the same reaction mixtures [21], we estimated that the reaction progress during cooling and heating corresponds to 3.4 min long isothermal heating at 40 °C. Therefore, the formal time interval between the two measurements was approximately 13.4 min.

### 5.6. Kinetic Analysis

Except for the DLS analysis, the methods used enable the transformation of measured signals to an approximation of the degree of conversion α.

For FTIR, we used absorbance at 1151 cm^−1^ to determine the conversion as follows:(5)$$\alpha_{t}=\frac{A_{0}-A_{t}}{A_{0}-A_{\infty }},$$
where *A*_∞_, *A*_0_, and *A_t_* are absorbances in 60 min, at the start and at time *t*, respectively.

From the NMR relaxometry data, the portion of component 2 was used in an analogous formula as follows:(6)$$\alpha_{t}=\frac{C2_{0}-C2_{t}}{C2_{0}-C2_{\infty }},$$
where *C*2_∞_ is the maximal content, *C*2_0_ is the minimal content, and *C*2*_t_* is the content of component 2 in time *t*.

The decrease in the melting point Δ*T* measured by DSC is related to the concentration and composition of the melting solution, thus it can also serve as a measure of conversion.
(7)$$\alpha_{t}=\frac{\Delta T_{0}-\Delta T_{t}}{\Delta T_{0}-\Delta T_{\infty }},$$
where Δ*T*_∞_ is the decrease of melting point at the end of reaction, Δ*T*_0_ at the reaction start and Δ*T_t_* in time *t*.

The obtained *α*(*t*) functions were fitted with a basic kinetic equation involving a simplified Sestak-Berggren reaction model [34] as follows:(8)$$\frac{d\alpha }{dt}=k\times {\left(1-\alpha \right)}^{n}\times \alpha^{m}$$

The rate constant *k* (at 40 °C) together with the parameters *m* and *n* related to the reaction model were determined with the Solver tool in MS Excel.

## Figures and Tables

**Figure 1 gels-08-00008-f001:**
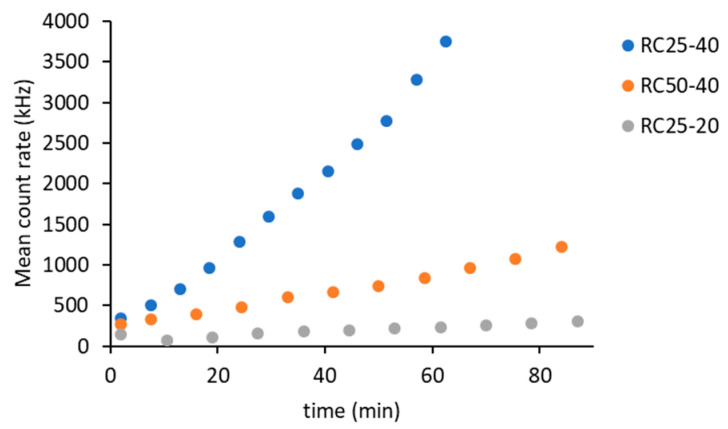
Time evolution of the scattered intensity measured for the studied RF mixtures.

**Figure 2 gels-08-00008-f002:**
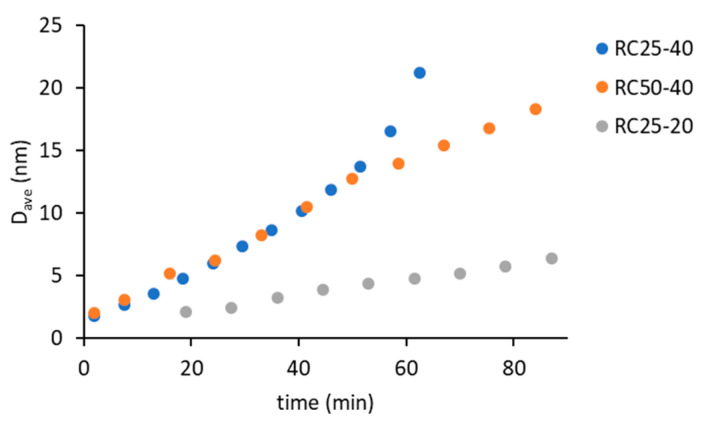
Dependence of average particle diameter in the studied RF mixtures.

**Figure 3 gels-08-00008-f003:**
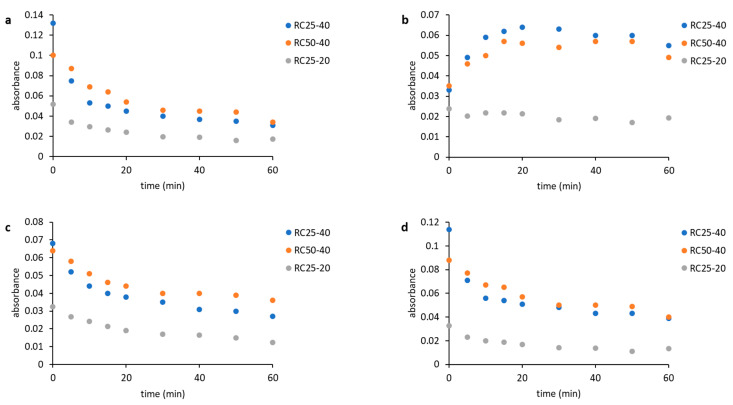
Time dependence of absorbances of studied RF mixtures, (**a**) 1151 cm^−1^, (**b**) 1114 cm^−1^, (**c**) 1057 cm^−1^, (**d**) 964 cm^−1^.

**Figure 4 gels-08-00008-f004:**
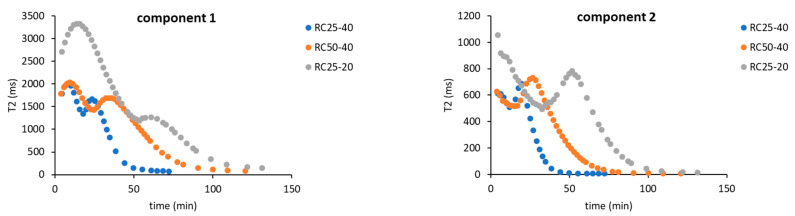
Relaxation times of two expected components during RF condensation.

**Figure 5 gels-08-00008-f005:**
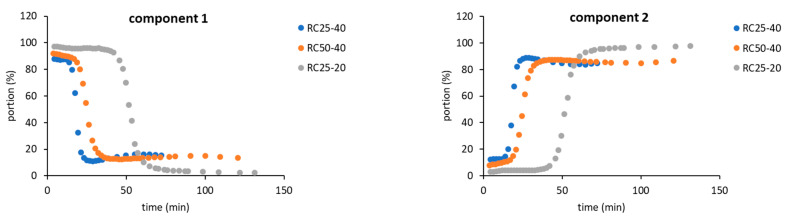
Content of two expected components during RF condensation.

**Figure 6 gels-08-00008-f006:**
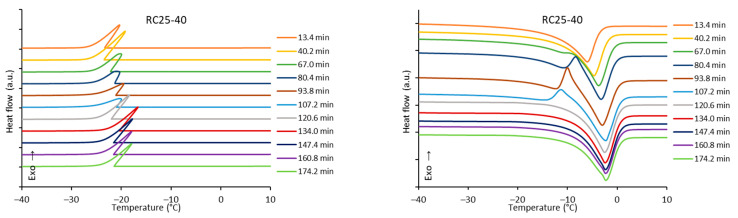
Development of the freezing (**left**) and melting (**right**) curves of reaction mixtures over time. The scale marks on heat flow axes represent 5 W/g for freezing and 1 W/g for melting curves.

**Figure 7 gels-08-00008-f007:**
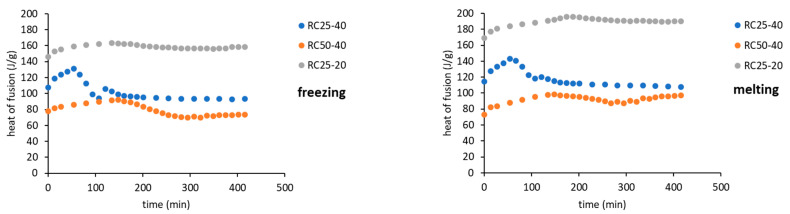
Peak areas (heat of fusion) during freezing and melting of reaction mixtures.

**Figure 8 gels-08-00008-f008:**
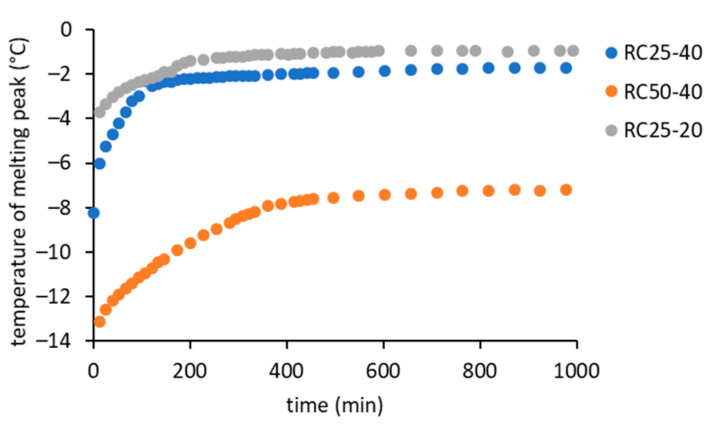
Increase of the melting point during the condensation of the studied RF mixtures.

**Figure 9 gels-08-00008-f009:**
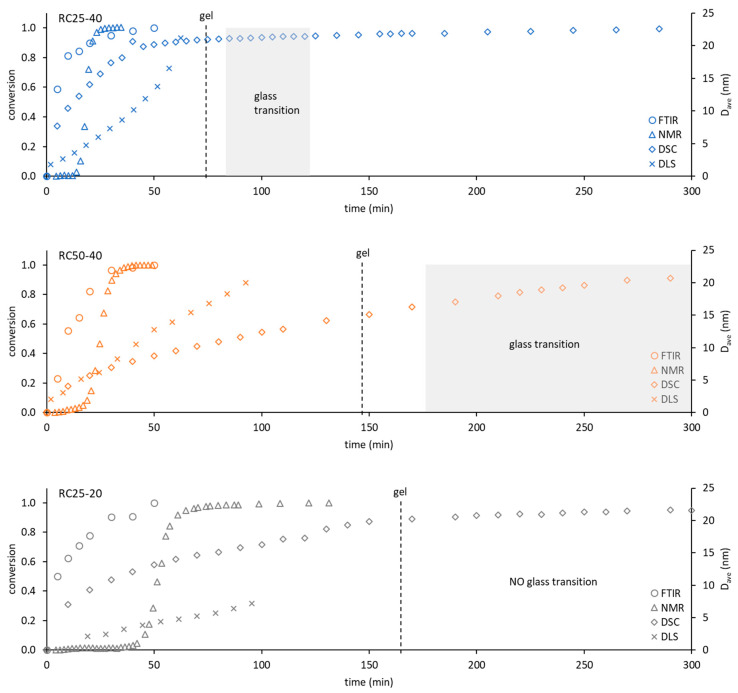
Comparison of the used techniques in terms of time dependence of conversion (FTIR, NMR, DSC—details of the calculation in the Section 5) or the particle diameter (DLS). The dashed line marked as ‘gel’ indicates the time of gelation (tilt test); the gray area corresponds to the time interval in which the glass transition could be detected by DSC.

**Table 1 gels-08-00008-t001:** Kinetic parameters based on different techniques.

Technique	Sample	*k* (1/min)	*n*	*m*
	RC25-40	0.3	1.5	0.1
FTIR	RC50-40	0.1	0.8	0.2
	RC25-20	0.1	1.6	0.0
	RC25-40	0.8	1.1	1.0
NMR	RC50-40	0.4	1.0	0.9
	RC25-20	0.4	1.3	1.0
	RC25-40	0.03	1.6	0.0
DSC	RC50-40	0.01	1.2	0.0
	RC25-20	0.02	1.6	0.0

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
