# Peer review of "Kinetics of Resorcinol-Formaldehyde Condensation—Comparison of Common Experimental Techniques"

_gels, 2021, doi:10.3390/gels8010008_

Round 1

Reviewer 1 Report

The manuscript entitled “Kinetics of resorcinol-formaldehyde condensation - comparison of common experimental techniques” is an excellent work. Some points need attention and clarification:

  1. What is the importance of knowing the kinetic parameters?
  2. Figures 4 and 5 show the evolution of relaxation times T2 of both components with reaction time and their abundance, respectively. However, explaining the meaning of both components is not easy to understand. So, please, present the interpretation of each component.
  3. Figure 6 represents heating flow over temperature. Confirm if the heating flow does not have a unit (e.g., Watt).
  4. There is not a conclusions section. However, it is important to conclude the main results according to the novelty of the work.

Reviewer 2 Report

This work presents the study on kinetics  of RF condensation with different catalyst content and reactant concentration by several analysis methods. The results were discussed in detail. The suggestion is listed as follow.

  1. Please clarify that the particle size measured by DLS was the result of primary particles or aggregation.  
  2. As discussed in section of DSC, the release of water was effected by glass transition of polymer. How to obtain the glass transition temperature of polymer?
  3. The influence factors on sol-gel were complex. The work discusses the effects of catalyst content, reactant concentration and reaction time. The other factors such as types of solvent and catalyst could be investigated by future research.

Round 2

Reviewer 1 Report

The new version is approved.